# Noise-Robust Audio-Visual Speech-Driven Body Language Synthesis

## Abstract

With the continuous advancement of video generation, researchers have achieved speech-driven body language synthesis, such as co-speech gestures. However, due to the lack of paired data for visual speech (i.e., lip movements) and body languages, existing methods typically rely solely on audio-only speech, which struggles to correctly synthesize target results in noisy environments. To overcome this limitation, we propose an **A**udio-**V**isual **S**peech-**D**riven **S**ynthesis (**AV-SDS**) method tailored for body language synthesis, aiming for robust synthesis even under noisy conditions. Given that each body language modality data has its corresponding audio speech, AV-SDS adopts a two-stage synthesis framework based on speech discrete units, consisting of the `AV-S2UM` and `Unit2X` modules. It uses speech discrete units as carriers to construct a direct mapping from audio-visual speech to each body language. Considering the distinct characteristics of different body languages, AV-SDS can be implemented based on semantic and acoustic discrete units, respectively, to achieve high-semantic and high-rhythm body language synthesis. Experimental results demonstrate that our AV-SDS achieves superior performance in synthesizing multiple body language modalities in noisy environments, delivering noise-robust body language synthesis. For samples and further information, please visit demo page at `https://av-sds.github.io/`.

## 1 Introduction

Recent years have witnessed great advancements in video generation (Tian et al., 2024; Richard et al., 2021), and researchers have successfully achieved various forms of speech-driven body language synthesis (Liu et al., 2023), including talking head generation (Prajwal et al., 2020; 2022), co-gesture generation (Liu et al., 2022b;a; Yang et al., 2023b), 3D facial animation (Richard et al., 2021; Fan et al., 2022; Xing et al., 2023), etc. Despite significant progress in these fields, existing methods are limited to employing audio-only speech for synthesis, as shown in Figure 1. However, in some complex environments such as construction sites and plants (Ahmed & Gadelmoula, 2022), it is difficult to extract clear audio signal, especially where headsets are unavailable due to regulations of production safety.

To understand speech in noisy scenes, researchers often use visual speech assistance (Afouras et al., 2018b; Shi et al., 2022b). For example, (Wang & Zhu, 2021) proposed a vision-based human-machine communication framework through gesture recognition, and (Ray & Teizer, 2012) also confirmed the feasibility of real-time posture analysis used in workers' ergonomics training. However, in most speech-driven body language synthesis tasks, such as Cross-ID talking head synthesis (Wang et al., 2021; Liang et al., 2022), Cross-ID landmark generation (Hsu et al., 2022), 3D facial animation, and speech gesture synthesis, there is a paucity of paired data between visual speech and body languages (Daněček et al., 2022). This scarcity hinders the realization of audio-visual speech-driven body language synthesis models.

Since the audio speech modality has a large amount of data supporting self-supervised learning (Kahn et al., 2020; Zen et al., 2019), many researchers have successfully used speech discrete units to represent speech information. Inspired by direct speech-to-speech translation (Lee et al., 2021), and considering that each body language (including visual speech, i.e., lip movements) is aligned with audio and has ample training data, can we use discrete speech units extracted from audio

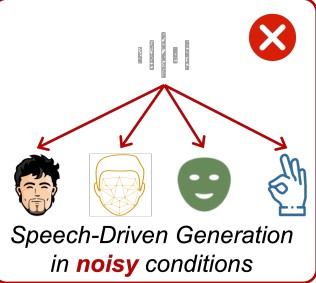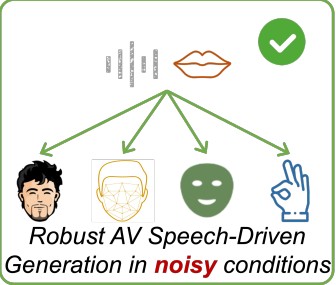

Figure 1: Overview of speech-driven multimodal synthesis tasks under noise-free and noisy conditions. The audio-only speech-driven approach is inadequate for synthesis in noisy environments. Conversely, the audio-visual speech-driven method can effective enhance the robustness of speech-driven synthesis in noisy settings. This paper focuses primarily on four tasks: Cross-ID talking head generation, Cross-ID landmark generation, 3D facial animation, and co-speech gesture generation.

speech self-supervised learning (SSL) models (Hsu et al., 2021; Zeghidour et al., 2021) as a bridge between visual speech and body language to address this challenge and achieve direct audiovisual speech-driven multimodal body language synthesis?

In this paper, we introduce a two-stage audio-visual speech-driven body language synthesis model (AV-SDS) based on speech discrete units, which includes two basic modules: `AV-S2UM` and `Unit2X`. The `AV-S2UM` module consists of an audio-visual speech encoder (Shi et al., 2022a) and several transposed convolutional layers, which can map the audiovisual speech to the corresponding discrete speech units. Subsequently, these discrete speech units are input into the `Unit2X` module to synthesize the corresponding multimodal body language data. In particular, to meet the characteristics of different body language modalities, the AV-SDS can be implemented based on different speech discrete units: an acoustic-centered model for co-speech gestures which focus more on the emotion and rhythm of speech (Loehr, 2007), implemented based on acoustic discrete units; and a semantic-centered model for modalities that focus on the semantic information of speech (talking head, facial landmark, and 3D face mesh), proposed based on semantic discrete units. Experiments show that the `Unit2X` module can successfully resynthesize various body languages from discrete speech units. Furthermore, experiments on various speech-driven body language synthesis tasks under different noise conditions demonstrate that our AV-SDS is capable of noise-robust speech-driven synthesis. The main contributions are as follows:

- We propose a novel two-stage, noise-robust, audio-visual speech-driven body language synthesis model (AV-SDS) based on discrete speech units.
- We propose `AV-S2UM` module, which excels in retaining speech information in noisy environments.
- We propose `Unit2X` module to synthesize body language data from discrete speech units, demonstrating the sufficiency of speech information in these units for body language synthesis.
- Our experiments confirm the robustness and effectiveness of AV-SDS across varying noise conditions, validating its potential for noise-robust speech-driven body language synthesis.

## 2 RELATED WORKS

### 2.1 SPEECH DRIVEN MULTI-MODAL BODY LANGUAGE SYNTHESIS

Body language (Liu et al., 2023) plays a pivotal role in facilitating effective communication and enhancing social interactions. Over the years, numerous researchers (Ye et al., 2023; Zhou et al., 2020) have dedicated their efforts to speech-driven body language synthesis, aiming to create digital avatars that seamlessly synchronize with spoken content. Leveraging advancements in generative technology, researchers have made notable progress, culminating in the development of both 2D talking head avatars (Zhou et al., 2020; Prajwal et al., 2020) and 3D facial animation (Richard et al., 2021; Xing et al., 2023), driven by audio and speech inputs. Moreover, recognizing the importance of comprehensive avatar technology, researchers (Yang et al., 2023b;c) have expanded their exploration into co-speech gesture synthesis, harnessing the rhythm and cadence of speech to imbue digital avatars with lifelike gestures.

However, existing methods can only achieve multimodal body language synthesis driven solely by audio speech, lacking robustness in noisy environments. To address this gap, we propose the first direct audio-visual speech driven multi-modal body language synthesis model, enabling speech-driven synthesis even in noisy conditions.

## 2.2 ROBUST AUDIO-VISUAL SPEECH LEARNING

Audio speech understanding (Baevski et al., 2020; Hsu et al., 2021) technology has advanced rapidly, effectively conveying speech content information. However, in noisy environments like outdoors, audio-only models often lack robustness and struggle to resist environmental noise interference. To address this, researchers (Afouras et al., 2018b) have begun exploring the use of visual speech (lip movements) to enhance speech understanding capabilities under noisy conditions. Some researchers (Afouras et al., 2018a;b) initially collected paired audio-visual speech corpus from TED and BBC for research. Subsequently, AV-HuBERT (Shi et al., 2022a) achieves noise-robust audio-visual speech recognition as well as lip reading, successfully recognizing speech content in noisy condition. Additionally, some (Gao & Grauman, 2021; Hsu et al., 2023) propose using visual speech to tackle the cocktail party problem, effectively addressing the challenge of distinguishing the active speaker track among multiple speakers.

However, in speech-driven multimodal generation tasks, many body language modalities (e.g., meshes and gestures) lack paired data with visual speech (Daněček et al., 2022), hindering robust audio-visual speech-driven synthesis. In this paper, we introduce a two-stage AV-SDS framework that utilizes discrete speech units as carriers to overcome this challenge of unpaired data.

## 2.3 SELF-SUPERVISED LEARNING IN SPEECH

Self-supervised learning methods (Baevski et al., 2020; Hsu et al., 2021; Zeghidour et al., 2021; Yang et al., 2023a) leverage unlabeled audio speech data to significantly enhance speech representation and improve the performance of various speech-related tasks. For instance, some researchers (Baevski et al., 2020; Hsu et al., 2021) employ a continuous alternation between unsupervised clustering and mask prediction to augment the contextual semantic representation. Later, some (Zeghidour et al., 2021; Yang et al., 2023a) integrate the RVQ (residual vector quantization) module to achieve a fine-grained representation of speech acoustic information. Building upon these advancements, researchers have proposed utilizing Speech SSL model to discretize speech, thereby exploring new capabilities of speech models. Specifically, Lee et al. (2021) employs semantic discrete units as a bridge to achieve direct speech-to-speech translation between speeches of different languages, while Jiang et al. (2023) employs acoustic discrete units to represent acoustic information of speech, enabling zero-shot TTS.

While speech discrete units hold promise in various speech-related tasks, their application in speech-driven multi-modal generation tasks remains largely unexplored. In this paper, we introduce a unit-based multi-modal generation module, `Unit2X`, which represents a pioneering effort in leveraging speech discrete unit information for multi-modal generation exploration.

## 3 AUDIO-VISUAL SPEECH DRIVEN SYNTHESIS

### 3.1 OVERVIEW

Audio-visual speech-driven multimodal synthesis aims to generate multimodal content $M = \{M_1, M_2, \cdots, M_N\}$ (e.g., meshes, talking heads, gestures, etc.) consistent with audio speech $A = \{A_1, A_2, \cdots, A_N\}$ and visual speech $V = \{V_1, V_2, \cdots, V_N\}$, where $N$ is the number of audio-visual speech frames. Due to the lack of extensive paired data between visual speech and body language modalities (e.g., meshes and co-speech gestures), it is challenging to train an audio-visual speech-driven body language synthesis model. In this context, as shown in Figure 2, the AV-SDS proposed in this paper aims to use speech self-supervised learning (Speech SSL) model to map speech paired with various modalities into the corresponding unified speech discrete unit space $U = \{U_1, U_2, \cdots, U_T\}$, where $T$ is the number of discrete speech units. By using discrete speech units as carriers, we achieve direct audio-visual speech-driven multimodal synthesis. Specifically, as described in Section 3.2, we adopt the `AV-S2UM` module to construct the mapping from audio-visual

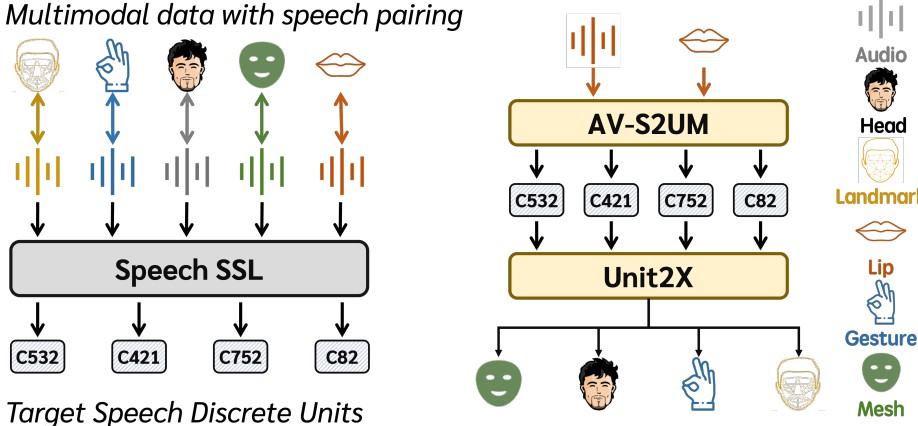

Figure 2: Illustration of **AV-SDS**. While there is no paired data between visual speech (lip movements) and most modalities (such as mesh and co-speech gestures), paired speech data exists for each modality in the speech-driven synthesis task. In this context, we leverage Speech SSL model (Hsu et al., 2021; Yang et al., 2023a) to convert the audio speech data from various modal pairs for the speech-driven generation task into corresponding speech discrete units, acting as a bridge between audio-visual speech and different modalities. As detailed in Section 3.2, we employ `AV-S2UM` to translate audio-visual speech into the target discrete speech units, followed by the `Unit2X` module introduced in Section 3.3 to synthesize the corresponding multi-modal data, thus achieving robust audio-visual speech-driven multimodal synthesis.

speech to unified speech discrete units. Subsequently, in Section 3.3, we introduce `Unit2X`, a unit-based multimodal generation framework, to reconstruct corresponding multimodal body language data from unified speech discrete units.

## 3.2 AUDIO-VISUAL SPEECH-TO-UNIT MAPPING SYSTEM

**Speech Discrete Units.** The information conveyed by speech can be broadly classified into two categories: semantic information and acoustic information. HuBERT (Hsu et al., 2021) employs multiple iterations of mask prediction and $\mathcal{K}$-means clustering to continuously enhance its ability to understand speech context, effectively extracting semantic discrete units that encapsulate speech semantics, denoted as $U^s$. Encodec (Défossez et al., 2022) and SoundStream (Yang et al., 2023a; Zeghidour et al., 2021) utilize the RVQ module for precise speech reconstruction, obtaining the acoustic discrete unit, denoted as $U^a$. In this paper, we utilize the semantic discrete unit $U^s$ for generating modalities closely associated with semantics, such as the talking head, facial landmarks, and the mesh of a 3D avatar. Conversely, the acoustic discrete unit $U^a$ is employed for generating co-speech gestures, which are more closely related to emotion and rhythm.

**AV-S2UM** In Figure 2, each visual speech $V$ is paired with its corresponding audio speech $A$. We implement the AV-S2U-Mapper model based on the AV-HuBERT model, pre-trained on a large dataset of paired audio-visual speech utterances. While it's possible to use the AV-HuBERT representation directly to obtain discrete units, we opt for the Speech SSL model, trained specifically on speech data, to ensure unified speech discrete units across different modalities. We can obtain the acoustic discrete units ($U^a_{\text{lip}}$) and semantic discrete units ($U^s_{\text{lip}}$) corresponding to the audio speech in $(A, V)$.

By leveraging the pre-trained AV-HuBERT model (Shi et al., 2022a), we embed the audio-visual speech into robust speech features $\mathbf{f}$, denoted as $\mathbf{f} = \text{AV-HuBERT}(A, V)$. Subsequently, we employ $n$ transposed convolutional layers to align these features with discrete unit sequences ($n = 1$ for semantic discrete units and $n = 3$ for acoustic discrete units). Each layer utilizes a kernel size ($K$) of 4, a stride ($S$) of 2, padding ($P$) of 1, and output padding ($Op$) of 1. The output size ($O$) of each transposed convolutional layer is calculated using the formula $O = ((I - 1) \times S + K - 2 \times P) + Op$.

The target distribution $p(U_t|\{U_i\}_{i=1}^{t-1}, (A, V))$ can be obtained with the robust speech feature $\mathbf{f}$:

$$p(U_t|\{U_i\}_{i=1}^{t-1}, (A, V)) = \text{AV-S2UM}(\mathbf{f}), \tag{1}$$

and the `AV-S2UM` module is trained using the cross-entropy loss:

$$L_{\text{AV-S2UM}} = -\sum_{t=1}^{N} \log p(U_t | \{U_i\}_{i=1}^{t-1}, (A, V)). \tag{2}$$

### 3.3 UNIT-BASED MULTI-MODAL SYNTHESIZER

With the Speech SSL model, the speech discrete unit $U_m$ corresponding to the audio speech $A$ paired with the multi-modal data $M$ can be derived as either the corresponding semantic discrete unit ($U_m^s$) or the acoustic discrete unit ($U_m^a$). In this subsection, we will elucidate how the `Unit2X` module synthesizes corresponding multi-modal data from semantic discrete units or acoustic discrete units.

**Synthesizer based on Semantic Units.** Drawing inspiration from Polyak et al. (2021), we employ a lookup table (LUT) to map these discrete units $U_m^s = \{U_1^s, \cdots, U_T^s\}$ to the corresponding speech representation $\mathbf{f}^a = \{\mathbf{f}_1^a, \cdots, \mathbf{f}_T^a\} = \text{LUT}(U_m^s)$. Subsequently, these speech representations $\mathbf{f}^a$ are inputted into various models to generate corresponding body language data.

For the acoustic discrete unit, we utilize 8 distinct units to represent various granular speech features at each speech frame, denoted as $U_m^a = \{U_{1,1}^a, \ldots, U_{1,8}^a, \ldots, U_{T,1}^a, \ldots, U_{T,8}^a\}$. We obtain the audio speech representation $\mathbf{f}^a$ by employing the pre-trained RVQ (Residual Vector Quantization) module within the Speech SSL model (Yang et al., 2023a) for acoustic units: $\mathbf{f}^a = \{\mathbf{f}_1^a, \cdots, \mathbf{f}_T^a\} = \text{RVQ}(U_m^a)$. Note that we keep the parameters of the RVQ module frozen during training.

**Unit2X: Unit-Based multi-modal synthesizer.** In the traditional speech-driven body language synthesis task, the audio encoder is utilized to encode the raw audio into the corresponding audio speech feature $\mathbf{f}^a$. However, in this work, we substitute the original audio encoder with the unit-based encoder to obtain the audio speech feature. Once we obtain the audio speech representation $\mathbf{f}^a$, we can apply the model used in the traditional speech-driven multi-modal synthesis task to generate corresponding multi-modal data $M$:

$$M = \text{Unit2X}(\mathbf{f}^a). \tag{3}$$

Here, we illustrate this process using the task of talking head generation as an example. The audio speech feature $\mathbf{f}^a$ is first input into the face decoder, where it is upsampled and combined with $\mathbf{f}^s$, the latter being extracted from randomly selected speaker reference frames and pose prior frames. This combination generates the final talking head. The discriminator $D$ consists of a series of convolutional blocks and is trained alternately with the generator $G$. The loss function employed during model training remains consistent with traditional speech-driven multi-modal generation methods. Specifically, for the task of talking head generation, we use GAN loss $L_G$, lip reconstruction loss $L_{lip}$, and synchronization loss $L_{sync}$ as the training objectives:

$$L_{\text{unit2x(head)}} = (1 - \lambda_{sync} - \lambda_{gen})L_{lip} + \lambda_{sync}L_{sync} + \lambda_{gen}L_G, \tag{4}$$

where $L_{\text{unit2x(head)}}$ is the training objective of `Unit2X` for talking head generation, and $\lambda_{sync} = 0.03$ and $\lambda_{gen} = 0.07$ as proposed by Prajwal et al. (2020).

## 4 EXPERIMENTS

### 4.1 DATASETS

The `AV-S2UM` module is trained on the LRS3 dataset (Afouras et al., 2018b). For the `Unit2X` model, we utilize the most commonly employed datasets for each modality: 29h training split of LRS2 (Petridis et al., 2018) for talking head generation, LRS3 (Afouras et al., 2018b) for facial landmark synthesis, VOCASET (Cudeiro et al., 2019) for 3D facial animation, and TED-GESTURE (Yoon et al., 2019) for co-speech gesture synthesis. The pre-trained Speech SSL models used in this paper to obtain speech discrete units are trained on Libri-Light (Kahn et al., 2020) and the TTS Corpus (Yang et al., 2023a), respectively. Following the methodology described in Shi et al. (2022a), we introduce noise into the audio speech by incorporating samples from the MUSAN dataset (Snyder et al., 2015).

Table 1: Comparison of synthesis quality for various body language modalities. **S2X** denotes direct synthesis from real speech, **Unit2S+S2X** indicates synthesizing speech from units followed by additional synthesis, and **Unit2X** refers to the direct body language generation from discrete units.

| Method | Talking Head | | | Mesh | | Landmark | Gesture |
|---|---|---|---|---|---|---|---|
| | LSE-C↑ | LSE-D↓ | FID↓ | SYNC.↑ | REAL.↑ | LMD↓ | FGD↓ |
| *Synthesize X-modality data from real speech.* | | | | | | | |
| S2X | 7.50 | 7.14 | 5.08 | 45.01 | 39.72 | 4.287 | 4.133 |
| *Synthesize X-modality data from speech discrete units.* | | | | | | | |
| Unit2S+S2X | 5.74 | 7.96 | 5.52 | 41.37 | 35.29 | 5.180 | 4.228 |
| Unit2X(Ours) | **7.34** | **7.54** | **5.14** | **46.83** | **43.28** | **4.718** | **3.976** |

## 4.2 IMPLEMENTATION DETAILS

To ensure consistency across different body language modalities, we resample the audio speech in all datasets to 16kHz in this paper. This allows us to extract unified speech discrete units to represent semantic or acoustic information. Specifically, we use the HuBERT BASE model (Hsu et al., 2021) to extract semantic discrete units and the 16kHz version of the hificodec model (Yang et al., 2023a) to extract acoustic discrete units.

To facilitate effective research and encourage broad adoption, we intentionally selected fundamental, widely applicable implementations for each modality. Specifically, we use Wav2Lip (Prajwal et al., 2020) for talking heads, GeneFace (Ye et al., 2023) for 3D landmarks, CodeTalker (Xing et al., 2023) for face meshes, and Tri-Modal (Yoon et al., 2020) for co-speech gestures to implement the corresponding speech-driven body language synthesis tasks. We believe that experiments on basic implementations of different body language modalities are sufficient to demonstrate the effectiveness of our approach, which can be integrated with any speech-driven body language synthesis model in the future. For additional details and evaluation metrics, please refer to Appendix B.

## 4.3 UNIT2X: UNIT-BASED BODY LANGUAGE SYNTHESIS

Polyak et al. (2021) demonstrated the feasibility of reconstructing corresponding speech from discrete units. Building on this, we attempt to synthesize different body language data from corresponding speech discrete units. As shown in Table 1, we compare the performance of various synthesis methods across different body language modalities. This demonstrates that speech discrete units can effectively replace the original speech as input for synthesizing corresponding body language data. For additional qualitative comparisons, please refer to Appendix B.

**Audio-Based vs. Unit-Based.** The audio-based method (`S2X`) directly extracts the corresponding embedding from the mel spectrum for body language modality synthesis, while the unit-based method (`Unit2X`) synthesizes from discrete units. As demonstrated in the experiments, across various body language modalities, the unit-based method achieves performance comparable to that of the audio-based method, affirming that speech discrete units can efficiently represent the speech information necessary to generate the corresponding body language data. Additionally, in scenarios with limited training data (e.g., mesh) or where cross-modal mapping is challenging to build (e.g., co-speech gesture), unit-based methods prove to be more effective than methods using raw speech as input. For instance, the FGD of `Unit2X` is 3.976, while that of `S2X` is 4.133. This demonstrates that speech discrete units can distinctly capture various semantic and acoustic information from speech, leading to superior performance in these challenging contexts.

**U2S+S2X vs. Unit2X.** To synthesize the body language data corresponding to the speech discrete unit, the simplest approach is to first re-synthesize the audio speech corresponding to the discrete unit, and then synthesize the corresponding body language data from the audio speech (i.e., `U2S+S2X`). However, this cascade method tends to accumulate errors, resulting in a significant decline in the correlation between the generated body language modality data and the audio speech compared to direct synthesis from the unit (the LSE-C of `Unit2X` is 7.34, while the LSE-C of `U2S+S2X` is only 5.74). This further underscores the importance of the unit-based body language synthesizer (`Unit2X`).

Table 2: Comparison of speech-driven synthesis performance across four different modalities (talking head, co-speech gesture, facial landmark, and mesh) under varying noise conditions. We present performance comparisons across different signal-to-noise ratios (SNRs) of SNR = {15, 5, -5, -15}. The cascade method with a dagger ($^\dagger$) employs the AV-S2UM+U2S+S2X cascade method. The mesh modality results are assessed against the generated mesh of clean audio speech-driven synthesis.

(a) Comparison of Talking Head Generation on LRS3. LSE-C and LSE-D in this table are evaluated between the generated talking head video and the clean audio speech.

| Method | LSE-C ↑ | | | | LSE-D↓ | | | | FID↓ | | | |
|---|---|---|---|---|---|---|---|---|---|---|---|---|
| | 15 | 5 | -5 | -15 | 15 | 5 | -5 | -15 | 15 | 5 | -5 | -15 |
| Wav2Lip | 5.63 | 4.65 | 3.15 | 2.05 | **8.19** | 8.49 | 8.93 | 9.35 | **5.88** | 6.42 | 7.45 | 8.73 |
| Cascade$^\dagger$ | 5.45 | 5.23 | 5.09 | 4.83 | 8.47 | 8.67 | 8.94 | 9.12 | 6.12 | 6.26 | 6.41 | 6.67 |
| AV-SDS | **5.72** | **5.59** | **5.41** | **5.12** | 8.32 | **8.45** | **8.51** | **8.68** | 5.97 | **6.12** | **6.23** | **6.37** |

(b) Comparison of 3D Mesh of Talking Head Generation on LRS3. The SYNC. and REAL. in this table are expressed as preference ratios compared to the results generated based on clean audio speech.

| Method | SYNC. ↑ | | | | REAL. ↓ | | | |
|---|---|---|---|---|---|---|---|---|
| | 15 | 5 | -5 | -15 | 15 | 5 | -5 | -15 |
| CodeTalker | **45.13** | 38.12 | 29.22 | 20.31 | 42.62 | 36.44 | 26.83 | 18.28 |
| Cascade$^\dagger$ | 38.78 | 35.94 | 32.13 | 28.89 | 36.38 | 35.04 | 33.71 | 31.66 |
| AV-SDS | 40.29 | **39.75** | **38.32** | **34.49** | **43.13** | **41.26** | **39.85** | **36.72** |

(c) Comparison of Landmark Generation on LRS3.  (d) Comparison of Gesture Generation on AV-GES.

| Method | LMD ↓ | | | | Method | FGD ↓ | | | |
|---|---|---|---|---|---|---|---|---|---|
| | 15 | 5 | -5 | -15 | | 15 | 5 | -5 | -15 |
| GeneFace | **5.115** | 5.357 | 5.927 | 6.356 | Tri-modal | **4.322** | 4.726 | 5.217 | 5.910 |
| Cascade$^\dagger$ | 5.384 | 5.579 | 5.637 | 5.845 | Cascade$^\dagger$ | 4.607 | 4.689 | 4.772 | 5.367 |
| AV-SDS | 5.272 | **5.304** | **5.328** | **5.523** | AV-SDS | 4.434 | **4.524** | **4.580** | **4.843** |

## 4.4 ROBUST AUDIO-VISUAL SPEECH DRIVEN MULTI-MODAL SYNTHESIS

To assess the performance of different speech-driven body language synthesis methods in noisy environments, we conducted validation across various body language modalities: **(1) For talking head and facial landmarks**, since there are no ground truth results in the cross-identity audio-visual speech-driven talking head or facial landmark generation task, we follow the experimental setting of Prajwal et al. (2020) and use talking heads and facial landmarks paired with audio speech as targets for audio-visual speech-driven synthesis. Talking heads are evaluated on the LRS2 dataset, while facial landmarks are evaluated on the LRS3 dataset. To further enhance the convincingness, we also present cross-identity audio-visual speech-driven talking head generation results in Section 4.6, validated solely through audio-visual synchronization (Prajwal et al., 2020). **(2) For 3D face mesh**, due to the absence of paired mesh and visual speech data for testing, we utilized the speech-driven method to generate 3D face mesh corresponding to clean audio speech on the LRS3 dataset, using these as reference videos for qualitative comparison. **(3) For co-speech gesture**, we curated a test set comprising audio-visual speech and co-speech gesture paired data by re-collecting and processing the original video clips from the TED-GESTURE dataset as described by Afouras et al. (2018b). This Audio-Visual TED-GESTURE (AV-GES) test dataset included 565 utterances in total. In Table 2, we present the experimental results across various tasks, demonstrating the effectiveness of our proposed AV-SDS method under noisy conditions.

**Noise-Robust Speech-Driven Synthesis** Traditional audio-only methods exhibit significant performance degradation in noisy environments. For instance, in Table 2d, the audio-only speech-driven method (Tri-modal) experiences a notable decrease in performance, dropping by 1.588 from 4.322 to 5.910 as SNR decreases from 15 to -15. In contrast, our proposed AV-SDS, driven by audio-visual speech, only experiences a slight decrease of 0.409 from 4.434 to 4.843. This demonstrates that AV-SDS achieves noise-robust speech-driven body language synthesis and is more robust to noise.

Table 3: Comparison of speech enhancement performance under different noise conditions on LRS3. We re-evaluated the LSE metrics following Prajwal et al. (2020), and reproduced the results of ReVISE based on the `AV-S2UM`($U^s$) module and unit-based vocoder (Polyak et al., 2021).

| Method | WER(%) ↓ | | | | LSE-C ↑ | | | | LSE-D ↓ | | | | MOS ↑ |
|--------|------|------|------|------|------|------|------|------|------|------|------|------|------|
| | 15 | 5 | -5 | -15 | 15 | 5 | -5 | -15 | 15 | 5 | -5 | -15 | Avg. |
| Inp.Audio | 7.8 | 17.8 | 63.9 | 87.9 | 6.72 | 6.68 | 3.38 | 2.05 | 7.58 | 8.12 | 10.82 | 11.93 | 2.88±0.17 |
| Resynthesis | 10.2 | 19.9 | 83.5 | 97.7 | 6.64 | 6.53 | 3.19 | 1.82 | 7.65 | 8.29 | 11.04 | 12.14 | 2.81±0.16 |
| Demucs | 6.9 | 15.1 | 48.0 | 81.3 | 6.98 | 6.89 | 3.87 | 2.21 | 7.38 | 7.83 | 10.12 | 11.79 | 3.22±0.15 |
| VisualVoice | 6.6 | 8.8 | 23.4 | 58.0 | 7.03 | 6.75 | 5.70 | 4.81 | 7.34 | 7.61 | 8.57 | 9.79 | 3.52±0.12 |
| ReVISE | 9.4 | 9.7 | **11.7** | **20.5** | 6.63 | 6.59 | 6.41 | 5.79 | 7.49 | 7.56 | 7.78 | 8.46 | 3.40±0.13 |
| AV-S2UM($U^a$) | **5.8** | **6.5** | 13.6 | 46.0 | **7.25** | **7.14** | **7.06** | **5.85** | **7.15** | **7.32** | **7.51** | **8.30** | **3.79±0.12** |

Particularly under harsh noise conditions (SNR= $\{5, -5, -15\}$), AV-SDS achieves superior performance in speech-driven body language synthesis, preserving more information from the speech and yielding more reliable body language synthesis. It's worth noting that since the `AV-S2UM` model is trained solely on LRS3, the experiments on co-speech gesture in Table 2d represent zero-shot scenarios for the `AV-S2UM` module. Even in such conditions, AV-SDS demonstrates better performance under noisy conditions, underscoring the significance of audio-visual speech-driven synthesis.

**Direct System vs. Cascade System.** While the cascade approach using the three modules `AV-S2UM+U2S+S2X` can achieve a certain degree of noise-resistant audio-visual speech-driven synthesis, the accumulation of errors from multiple module cascades hampers its synthesis performance under varying noise conditions compared to AV-SDS. Particularly, robust audio-visual speech understanding in noisy environments is significantly challenging. The errors generated by each module under noisy conditions are non-negligible and significantly impact the final outcome. For instance, in the results for SNR= $-15$ in Table 2d, the FGD of the cascade method is 0.524 lower than that of AV-SDS. Therefore, minimizing the number of cascade layers is crucial for speech-driven synthesis tasks in noisy environments.

### 4.5 AV-S2UM: PRESERVATION OF SPEECH INFORMATION IN NOISY AUDIO.

Based on visual speech, `AV-S2UM` effectively preserves speech information in noisy environments and can reconstruct corresponding audio speech using speech discrete units. As shown in Table 3, we compared the speech enhancement performance to evaluate the ability of various models to retain speech information under different noise conditions. Among these models, Demucs (Defossez et al., 2020) represents an audio-only approach, VisualVoice (Gao & Grauman, 2021) represents a naive audio-visual method, ReVISE (Hsu et al., 2023) relies on semantic discrete units (i.e., `AV-S2UM`($U^s$)), and `AV-S2UM`($U^a$) refers to the model that relies on acoustic discrete units.

**Audio-Only vs. Audio-Visual.** In a noisy environment, models that rely solely on audio-speech are unable to resist noise interference and suffer significant loss of speech information. The Word Error Rate (WER) for the audio-only method (Demucs) at SNR=-15 is 81.3%, showing only a 6.6% improvement from 87.9% for Inp.Audio. However, audio-visual speech-based methods use visual speech as auxiliary information to help models resist noise interference. VisualVoice, for example, maintains a WER of 58.0% at SNR=-15, which is 23.3% better than the audio-only method, demonstrating the importance of visual speech for audio understanding in noisy environments.

**Naive Audio-Visual Method vs. Unit-Based Audio-Visual Method.** Traditional visually guided speech enhancement methods only rely on a limited amount of audio-visual speech pairing data for training, making it difficult to achieve high-fidelity audio reconstruction. However, methods based on discrete units, such as ReVISE and our `AV-S2UM`($U^a$), train the `AV-S2UM` module and the corresponding vocoder separately on large-scale audio-visual speech and massive audio speech, achieving more effective audio information retention. Under the conditions of SNR = -5 and SNR = -15, the WER of unit-based ReVISE is 12% and 37.5% better than that of end-to-end VisualVoice, respectively. This experiment demonstrates that the two-stage method, which has more training data, can retain more original audio information compared to the end-to-end model, which can only use extremely limited paired data.

**Semantic Discrete Unit vs Acoustic Discrete Unit.** ReVISE employs semantic discrete units to retain semantic speech information while disregarding acoustic elements such as timbre, emotion, and rhythm (the speaker's timbre in the ReVISE speech output differs from the original timbre). In contrast, `AV-S2UM` ($U^a$) advances by utilizing acoustic discrete units as intermediaries to connect visual speech with high-fidelity audio speech. `AV-S2UM` ($U^a$) not only preserves speech semantic information but also retains fine-grained acoustic details, including emotion, timbre, and rhythm. It excels in audio-visual synchronization (LSE-C and LSE-D are the best among all noise conditions), showcasing its applicability to body language modalities closely related to emotion and rhythm, such as co-speech gestures. Notably, although ReVISE cannot retain the acoustic speech information, it maintains an excellent WER even under high noise conditions (WER of 20.5% at SNR=-15). It demonstrates that the `AV-S2UM` ($U^s$) module based on semantic discrete units can effectively preserve semantic speech information in noisy environments, making it well-suited for synthesizing body language modalities related to speech semantics.

## 4.6 Audio-Visual Speech-Driven vs. Video-Driven.

Although we use visual speech (i.e., lip movements) to facilitate audio-visual speech-driven body language synthesis, our AV-SDS differs significantly from the video-driven body language synthesis methods (Pang et al., 2023). Typically, video-driven methods generate the target individual's expressions and gestures based on a reference video. In contrast, our approach relies solely on speech information extracted from audiovisual speech to ensure that the generated results are consistent with the driving speech, without considering other visual information from the reference video. Given the difficulty in distinguishing speaker identity details from facial movements, results generated by video-driven methods (Yin et al., 2022) often retain speaker identity attributes (such as face shape and makeup) from the driving video. Additionally, we present lip sync metrics for cross-identity driven talking head synthesis in Table 4. Due to the lack of speech-related supervision, video-driven methods struggle to maintain a high level of lip sync during the synthesis process, only simulating the expressions of the driving video to a limited extent. Consequently, the generated video cannot effectively convey the corresponding speech content. Notably, even under extremely strong noise interference conditions (SNR$= -5$), AV-SDS outperforms video-driven method (DPE), demonstrating the relevance of our method for the task of audio-visual speech-driven body language synthesis.

Table 4: Comparison of cross-id talking head generation results using different modality driving methods on LRS3 at SNR=-5. The LSE metrics (LSE-C and LSE-D) are evaluated between the generated talking head video and the clean audio speech. AO-Speech: Audio-Only Speech.

| Method | Driven | LSE-C↑ | LSE-D↓ |
|---------|-----------|--------|--------|
| Wav2Lip | AO-Speech | 2.467 | 9.466 |
| DPE | Video | 3.620 | 9.824 |
| AV-SDS | AV-Speech | **4.447** | **9.242** |

## 5 Conclusion

Speech-driven body language synthesis aims to create intelligent digital humans that align with audio speech. However, due to the lack of paired data for visual speech and body language modalities, existing methods can rely on audio-only speech, which struggles to produce accurate results under noisy conditions. To address this issue, we propose the first direct audio-visual speech-driven multi-modal synthesis framework, AV-SDS. This framework employs speech discrete units as an intermediate carrier in a two-stage approach to bridge audio-visual speech and various body language modalities. Firstly, `AV-S2UM` maps audio-visual speech to unified discrete units. Then, `Unit2X` synthesizes various multi-modal body language data from these units. We introduce `Unit2X`, the first multi-modal body language synthesis model based on speech discrete units, and demonstrate the feasibility of using speech discrete units instead of raw audio speech for body language synthesis. Additionally, we propose two different implementations based on semantic discrete units and acoustic discrete units for semantically related and rhythm-related body language modalities, respectively. In various speech-driven multi-modal body language synthesis tasks, our AV-SDS achieves state-of-the-art performance under different noise conditions, confirming its effectiveness in noisy environments.

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

## A  DATASETS

**TTS Corpus (Yang et al., 2023a).**   The TTS Corpus integrates various public datasets, including LibriTTS (Zen et al., 2019) and VCTK (Veaux et al., 2016), encompassing 1,000 hours of high-fidelity English speeches. Importantly, all speech segments within the corpus have been meticulously confirmed to be free of discernible background noise. The 16kHz audio speech codec (Yang et al., 2023a) employed in this work is pre-trained on this corpus. This codec, rooted in the high-fidelity characteristics of the corpus, plays a crucial role in achieving the desired performance and fidelity in various speech-related tasks undertaken in this work.

**LRS2 (Afouras et al., 2018a) and LRS3 (Afouras et al., 2018b).**   LRS2 and LRS3 stand out as the most expansive publicly accessible lip-reading dataset at the sentence level, boasting over 229/443 hours of video content sourced from BBC and TEDx talks. In our experiments, we harnessed the train set to extract facial landmarks, following the methodology outlined by Ye et al. (2023).

In this paper, LRS3 serves as a crucial resource for evaluating the performance across various audio-visual speech-driven synthesis tasks for many body language modalities, such as talking head, facial landmarks and 3d mesh.

**VOCASET (Cudeiro et al., 2019).**   VOCASET consists of 480 paired audio-visual sequences recorded from 12 subjects. The facial motion is captured at 60fps, lasting approximately 4 seconds each. Each 3D face mesh is registered to the FLAME (Li et al., 2017) topology, featuring 5023 vertices. To ensure fair comparisons, we utilize the same training (VOCA-Train), validation (VOCA-Val), and testing (VOCA-Test) splits as VOCA (Cudeiro et al., 2019).

**TED GESTURE (Yoon et al., 2019).**   The TED Gesture Dataset encompasses a substantial volume of paired audio-visual sequences derived from TED talks, offering both a sizable dataset for investigating the intricate relationship between speech and gestures. Covering a diverse range of topics, TED talks feature thousands of unique speakers sharing their individual ideas and stories, capturing a broad spectrum of speech content.

**MUSAN (Snyder et al., 2015).**   In this paper, we randomly selected audio samples from MUSAN datasets to introduce background noise to the speech content. MUSAN (Snyder et al., 2015) consists of music, speech, and babble noise. Following the approach of Shi et al. (2022a), we used the audio samples from MUSAN to add noise to the speech.

## B  MORE IMPLEMENTATION DETAILS

### B.1  TRAINING DETAILS

For the training of `AV-S2UM` module, we loaded the publicly available pretrained weights of AV-HuBERT (Shi et al., 2022a) and fine-tuned the model over a total of 45,000 steps. During the first 5,000 steps, we exclusively trained the decoder by freezing the encoder. Afterward, we unfroze the encoder and trained the entire model together. The learning rate was adjusted using a tri-stage LR scheduling strategy with specific phases set at (10%, 20%, 70%) and a peak learning rate of 6e-5. The training was conducted on one V100 GPUs. We utilized the Adam optimizer with parameters set to (0.9, 0.98).

For the training of `Unit2X` module, we strictly follow the training details of each body language modality for training, and all models are trained on a single V100 GPU.

Table 5: Qualitative comparison of body language synthesis performance across different methods.

| Method | Talking Head | | Landmark | Gesture |
|---|---|---|---|---|
| | Qual. | Sync. | Sync. | Sync. |
| S2X | 4.12±0.09 | 4.09±0.13 | 3.76±0.18 | 3.88±0.15 |
| Ground Truth | 4.33±0.12 | 4.15±0.09 | 3.95±0.15 | 4.03±0.12 |
| U2S+S2X | 4.06±0.15 | 3.87±0.15 | 3.65±0.18 | 3.76±0.18 |
| Unit2X(ours) | **4.09±0.12** | **3.98±0.16** | **3.68±0.16** | **3.92±0.16** |

Table 6: Qualitative comparison for speech enhancement under various noise conditions.

| Method | SNR=15 | SNR=5 | SNR=-5 | SNR=-15 | Mean |
|---|---|---|---|---|---|
| Inp.Audio | 3.82±0.11 | 3.53±0.14 | 2.33±0.20 | 1.83±0.21 | 2.88±0.17 |
| Resynthesis | 3.74±0.09 | 3.46±0.17 | 2.26±0.19 | 1.78±0.20 | 2.81±0.16 |
| Demucs | 3.97±0.10 | 3.62±0.13 | 2.92±0.17 | 2.35±0.18 | 3.22±0.15 |
| VisualVoice | 4.01±0.08 | 3.77±0.10 | 3.42±0.13 | 2.87±0.16 | 3.52±0.12 |
| ReVISE | 3.52±0.09 | 3.42±0.12 | 3.37±0.15 | **3.29±0.15** | 3.40±0.13 |
| AV-S2UM($U^a$) | **4.25±0.07** | **4.05±0.09** | **3.68±0.12** | 3.18±0.18 | **3.79±0.12** |

## B.2 METRICS

**Unit-Based Body-language Synthesis.** (1) For lip movements, we employ LSE-C and LSE-D (Prajwal et al., 2020) as evaluation metrics to assess the synchronization between audio speech and lip movements. In the context of talking heads, we used FID (Heusel et al., 2017) to assess the dissimilarity between the generated images and the real images. (2) For facial landmarks, the facial landmark distance (LMD) (Chen et al., 2018) is used to measure the distance between the generated landmarks and ground truth landmarks. (3) For mesh, we conduct A/B testing to evaluate the authenticity (Real.) and synchronicity (Sync.) of various mesh synthesis methods. Similar to Xing et al. (2023), the evaluation is based on the percentage of samples with a higher user preference than ground truth (GT) videos. (4) For co-speech gesture, we employ FGD (Fused Gaussian Distance) as metrics. FGD (Yoon et al., 2020) measures the distribution disparity between generated output and ground truth with a pre-trained autoencoder.

**Speech Enhancement.** For the speech enhancement task, we employ the ASR model (Ott et al., 2019) to transcribe the denoised speech, using Word Error Rate (WER) as a metric to assess content accuracy. Additionally, to evaluate the synchronization between the denoised speech and the talking head video, LSE-C and LSE-D are adopted to assess lip synchronization.

## C  QUALITATIVE EXPERIMENTS

We performed a manual evaluation of all generated results, appraising qualitative outcomes through the Mean Opinion Score (MOS) methodology. Each sample was randomly presented to 15 participants for scoring. The composite MOS was subsequently calculated by averaging scores across the relevant dimensions. Each dimension was independently rated on a scale of 1 (lowest) to 5 (highest). Kindly visit the demo page (https://av-sds.github.io/) to view the corresponding generation results. The comprehensive MOS evaluation details for each task are outlined below:

**Unit-Based Body Language Synthesis.** In Table 5, we present a qualitative comparison among multiple methods on different body language modalities. For talking heads, we evaluated image quality (Qual.) and lip synchronization (Sync.). In the case of facial landmarks and co-speech gestures, the focus was on evaluating the synchronization (Sync.) of audio speech and bodyity data.

As the evaluation metric for the 3D mesh modality depends on manual assessment, we refrained from conducting further experiments on this particular modality.

**Speech Enhancement.** For the visually guided high-fidelity speech denoising task, we evaluated denoising outcomes across various noise conditions (SNR= $\{15, 5, -5, -15\}$), as depicted in Table 6. It is important to highlight that ReVISE consistently receives low subjective scores due to its inability to reconstruct the timbre of the corresponding speech. Notably, when SNR= $-15$, owing to its robust semantic reconstruction capability, ReVISE obtained the highest MOS. However, in other instances, AV-S2UM($U^a$) demonstrated superior high-fidelity speech noise reduction results, earning top ratings.

## D  LIMITATION

This paper verifies only a limited range of body language modalities (talking head, mesh, co-speech gesture, and 3D landmark). However, we believe these modalities are sufficient to demonstrate the effectiveness of our method. In the future, we will also validate it on additional modalities, such as listener responses and others.

## E  ETHICAL DISCUSSION

The task studied in this paper involves the field of virtual human synthesis, which carries a certain risk of video forgery. However, since the focus of this paper is not on the authenticity of the synthesis but on the robustness of the voice-driven body movement synthesis task in a noisy environment, this concern is not particularly serious.

