# OpenReview forum: "Noise-Robust Audio-Visual Speech-Driven Body Language Synthesis"
_ICLR.cc/2025/Conference — ICLR 2025 Conference Withdrawn Submission_

### Official Review · Reviewer_3rjf · 2024-10-30

**Soundness:** 2
**Presentation:** 1
**Contribution:** 2
**Rating:** 5
**Confidence:** 4

**Summary:**

This work addresses scenarios where paired data for visual speech (lip movements) and body language may be lacking, and the method may need to perform body language synthesis in noisy environments. In this context, the proposed Audio-Visual Speech-Driven Synthesis (AV-SDS) method utilizes a two-stage framework (AV-S2UM and Unit2X modules) based on speech discrete units, mapping audio-visual speech to body language. It leverages both semantic and acoustic units to achieve high-quality body language synthesis. Experimental results demonstrate that AV-SDS performs exceptionally well in synthesizing diverse body language modalities, maintaining effectiveness in noisy conditions.

**Strengths:**

-The experimental analysis is adequate, with controls implemented for various scenarios such as audio-based vs. uni-based, audio-only vs. audio-visual.
- The experimental results support the research question by showing that discrete speech units can effectively replace original speech as input to synthesize corresponding body language data.
- Qualitative results are convincing: https://av-sds.github.io/

**Weaknesses:**

-The necessity of Fig. 1 is questionable, as the situation is clearly explained in the text and seems relatively straightforward. Instead of Fig. 1, the methodology section could be extended.
-  In the contributions list, the first three bullet points could be merged. The method is described as novel, though several components rely on existing models. The novelty here seems to come from integrating various modules to accomplish the task.
- The method lacks a detailed description and requires a strong understanding of the leveraged techniques. Including a descriptive figure, at least in the appendix, would enhance clarity and provide valuable insights into the implementation.
- What distinguishes your UNIT2X approach from Prajwal et al., 2020?
- How does AV-S2UM differ from AV-HuBERT? Is it primarily the convolutional layers?

**Questions:**

- What distinguishes your UNIT2X approach from Prajwal et al., 2020?
- How does AV-S2UM differ from AV-HuBERT? Is it primarily the convolutional layers?

---

### Official Review · Reviewer_Aw7M · 2024-11-01

**Soundness:** 3
**Presentation:** 2
**Contribution:** 2
**Rating:** 3
**Confidence:** 4

**Summary:**

In this paper, the authors propose utilizing both audio and video speech to synthesize body language synthesis, which could achieve more  noise robustness compared to audio-only synthesis systems. Since there is no (or limited) paired data of video speech and body language, they implement a two-staged approach. Firstly, they design an audio-visual model to transform the audio-visual speech data into the discrete speech units. Then these discrete speech units are adopted for follow-up body language generation. Through this approach, the authors show that video speech is mainly used for enhancing noise robustness without leaking facial information for body language.

**Strengths:**

The paper is well-motivated: adding video speech to enhance the noise robustness of body language synthesis, whose validity has been widely proved by literature in speech domains. The experimental results also showcase the benefits by achieving better performances (under low SNRs) than baselines with various tasks.

**Weaknesses:**

1. My biggest concern is whether the proposed system is useful in realistic scenario. If we are able to access the lip movement, we already have the talking face. Why do we need to synthesize the talking face in such a scenario? Additionally, landmark, 3D facial, mesh can also be estimated by the video speech using pose/mesh estimation techniques. I truly understand the authors' motivation that only speech information in video speech is utilized. However, in real-world, why do we need such a system requires further clarifications.

2. Another concern (which may be minor) is that when introducing the new audio-visual speech data to train the system, the new speech data is also implicitly introduced. Is such fairness of training data between AV-SDS and baselines considered in the experiments? I expect that even under the fair setup, AV-SDS will still perform better than baselines with low SNRs, but how about the clean scenario.

3. The development of methodology in Section 3 is not well explained, which may also prevent potential readers from understanding the whole pipeline easily. (1) It is better to explicitly explain the formats / data shapes for inputs and outputs, especially for U. From my understanding, this unified speech discrete unit U is cluster number, thus scalar. Please correct me if I misunderstand. And please clarify the number of U in the vocabulary. (2) It is a little difficult to understand how the acoustic discrete units and semantic discrete units are extracted. So are there two AV-S2UM which are used to extract the above two Us, respectively?   (3) Can you explain more about how to implement the lookup table in section 3.3? To sum up, I think an illustration of model design, including submodules, inputs and outputs will be helpful for understanding the methodology, perhaps in Appendix if there is no space.

**Questions:**

See the above.

A minor question is which noise type is considered in the experiments. Normally, in speech domains, many different noises should be considered, e.g. bubble, natural noise, white noise, musical accompaniment, etc.

---

### Official Review · Reviewer_x6b6 · 2024-11-01

**Soundness:** 3
**Presentation:** 2
**Contribution:** 3
**Rating:** 3
**Confidence:** 4

**Summary:**

The goal of this work is to map speech to a set of discrete units such that the vocabulary of these units is consistent for driving a number of outputs, which include talking faces and speech accompanying manual gestures.  Furthermore, these outputs can take many forms, including video, facial landmarks, and 3D meshes.  At a high-level, speech is mapped to a HuBERT-like representation, which is then subsequently mapped to the respective output type.  The results show the approach is effective, even for high-levels of noise (up to -15db SNR tested).

**Strengths:**

Generating speech accompany information (visual speech and manual gestures) is an important and challenging problem for the creation of digital virtual characters.

The work makes effective use of many open datasets for different aspects of the generation.

Most of the results are compelling.

Evaluation considers both objective (automated) testing, and subjective (qualitative) testing, which is essential for assessing the quality of these generative models.

**Weaknesses:**

There is a conflation of terms when considering modalities.  For example, in the caption of Figure 2 you mention “modalities” and then give two examples:  mesh and co-speech gesture.  The context of “modality” for these two examples is different.  There is the modality of the data, which could be video, 3D mesh, 2D landmarks, etc., and there is the communication modality, which could be acoustic speech, visual speech, co-speech gesture.  Throughout the paper I found this conflation confusing.  Co-speech gesture can be represented as video, landmarks, or a 3D mesh.  Likewise can the visual modality of speech.

With a lot of these systems, the devil is in the detail.  Especially for a system like that described here that pulls together many models of different form.  The paper does not specifically state that code will be released.  It would significantly aid reproducibility if code were provided with the publication of the paper.

The motivating examples for the work are somewhat weak. Why do we need body language synthesis for constructions sites?  This seems somewhat niche.  I accept the problem is important and challenging, just that it is not well motivated here.

Some of the terminology is not clear.  For example, U^{a} represents discrete units for speech acoustic properties, and U^{s} represents discrete units for semantics.  Then in Equation (1) the probability distribution over U_{t} is discussed.  It is not clear what U_{t} is — is it both U^{a} and U^{s} individually, is a combination of the two, or is a combination of say U^{a} and U^{a}_{lip}?  There are other questions around this, see Questions, but this is a general problem I found with the paper.

Also to aid clarity, make sure all terms are defined.  For example, FGD is not defined in the main paper (it is in the Appendix).

**Questions:**

In the description of AV-S2UM — it is mentioned that the model used is based on AV-HuBERT pre-trained on a large dataset of paired  speech. In the next sentence it is stated that you “opt for the Speech SSL model, trained specifically on speech data, to ensure unified speech discrete units across different modalities”.  So is this just HuBERT and not AV-HuBERT since the model is pre-trained and based only on speech?

In the description of AV-S2UM — “We can obtain the acoustic discrete units (U^{a}_{lip}) and semantic discrete units (U^{s}_{lip})  corresponding to the audio speech in (A,V).”.  Obtain how?  Also, if the vocabulary of the discrete units is consistent across modalities, what is the difference between U^{a} and U^{a}_{lip}?

In Equation (1), is there a time index missing from A and V?

For the cross entropy loss in Equation (2), it looks like you are missing a term.

Why are eight levels of RVQ applied?  Is there any signal at this level of quantization?

Consider this sentence:  “The audio speech feature f^{a} is first input into the face decoder, where it is upsampled and combined with f^{s}, the latter being extracted from randomly selected speaker reference frames and pose prior frames.  How are these combined (added together?  concatenated?)?  What does ”randomly selected speaker reference frames and pose prior frames“ mean — individual frames are randomly selected?  A sequence is randomly selected?  How does the selection affect the generated sequence?

In Table 1 I find the heading confusing.  For example, does “Mesh” represent a facial mesh, a body face, both?  Likewise for “Landmarks”.  A talking head can be represented as landmarks, as a mesh, and as a video sequence.  Likewise for “Gesture”.

In Table 1, the baseline S2X beats both Unit2S+S2X and Unit2X for “Talking Head” and LMD of “Landamrks”, but it is not bolded.  Why?

Unit2S+S2X performs poorly compared with S2X, which suggests a domain shift in the reconstructed speech compared with real speech.  Why not train a model for S2X where the speech is the output from Unit2S?  Would this go someway to reducing the domain shift?

There is an assumption that noise robustness comes from mapping to discrete units and then reconstructing from these units rather than mapping directly from noisy speech.  Something that is not clear to me is why?  Does the noise added to speech not also affect the mapping to discrete units?  I would have thought that it would.

I am surprised that the system is able to construct the lip shapes and timing of these shapes so well for speech in -15db of babble noise.  In the example provided, I cannot hear the speech of the talker so it surprises me that the model can ignore all of the distractor speech (which is louder than the target speech) and focus on the target speech.

*Suggestions*
In the abstract you refer to body gestures, which is a very broad term since head motion, speech accompanying manual gesture, and facial expression, etc. could all be termed body gestures.  Here I think you are referring to co-speech manual gestures, so it would help to be specific.  This is also problematic in Section 3.1 for example when you say "aims to generate multimodal content M = {M_1, M2, ... M3} (e.g., meshes, talking heads, gestures, etc.)".  Again, because the distinction between data and communication modalities is not clear, it is not clear if the index is over modalities or over a sequence (it is the latter but that is not obvious).

Be clear when you are discussing different forms of modality so it is clear if you are referring to a data modality or a communication modality.  These are conflated throughout the paper. Co-speech gestures is not an alternative modality to meshes.  Co-speech gestures can be represented as a skeleton, video, landmarks, meshes, and so on.  The data modalities are alternatives for one another.

Personally I prefer “discrete speech units” over “speech discrete units”.

That Unit2X is an adversarial model is kind of sprung on the reader just by mentioning the generator and the discriminator, although it is not actually stated anywhere that the models is an adversarial model.  Maybe highlight this point earlier when the model is introduced.

In Equation 4, the loss terms L_{lip}, L_{sync}, and L_{G} are not defined.

The mean opinion score for Table 3 is not discussed in the main text — neither the setup for the qualitative experiment, nor the score itself.

---

### Official Review · Reviewer_BVkh · 2024-11-03

**Soundness:** 3
**Presentation:** 3
**Contribution:** 3
**Rating:** 6
**Confidence:** 3

**Summary:**

The authors present the Audio-Visual Speech-Driven Synthesis (AV-SDS) model, aimed at synthesizing robust, noise-resistant body language from audio-visual speech. Unlike prior methods relying solely on audio, AV-SDS incorporates discrete speech units to bridge audio-visual inputs with body language synthesis. The model operates in two stages: AV-S2UM maps audio-visual speech to discrete units, and Unit2X synthesizes multimodal body language directly from these units. The work demonstrates strong experimental results, showing AV-SDS’s efficacy in maintaining synthesis quality under noisy conditions, particularly for tasks like gesture generation and lip synchronization.

**Strengths:**

- The use of discrete units in bridging audio-visual and body language modalities is innovative, with a unique two-stage approach that is versatile across different body language synthesis tasks.
- Experimental rigor is high, as shown through comprehensive tests on standard datasets and metrics that validate the method’s robustness in noisy environments.
- The methodology is explained clearly, with well-defined modules and their respective functions, though minor simplifications could further enhance readability.
- The paper has potential in applications requiring robust audio-visual synthesis, especially in noisy settings where traditional audio-only methods are unreliable.

**Weaknesses:**

- While effective, the approach may be seen as incremental, with AV-SDS combining and refining existing techniques rather than introducing fundamentally new concepts.
- Some formulas and technical terms could benefit from clearer explanations or visual aids, particularly for multi-stage processes and the role of discrete units.
- The approach’s reliance on paired data might limit its applicability in domains with less paired audio-visual training data, which could be acknowledged and discussed as a limitation.

**Questions:**

- How does error propagation between the AV-S2UM and Unit2X modules affect synthesis quality, particularly under varying noise conditions? Did you do some investigations in this direction, too?
- Could the AV-SDS framework be adapted or scaled to handle body language modalities beyond those tested, such as more complex gesture or emotion-based responses?

**Details Of Ethics Concerns:**

The utilized datasets seem to be publicly available. However, as speech is highly personal data it should be made sure that the usage of all datasets is allowed.

---

### Official Review · Reviewer_xSgP · 2024-11-04

**Soundness:** 3
**Presentation:** 2
**Contribution:** 2
**Rating:** 5
**Confidence:** 4

**Summary:**

The paper proposes AV-SDS, a body language synthesis framework by speech signals under noisy conditions. AV-SDS is a two-stage approach, where in the first stage the AV-S2UM module maps noisy audio-visual speech input to discrete units using speech SSL models that capture both semantic and acoustic information. In the second stage, the Unit2X module uses these discrete units to synthesize body language signals (gestures, facial movements, and posture). Experimental results demonstrate that AV-SDS outperforms baseline methods in noisy conditions.

**Strengths:**

1. Novel two-stage approach to improve body language synthesis quality in noisy environments.
2. Using speech to output multiple body language signals instead of one.

**Weaknesses:**

1. Clarity of the paper can be improved, especially in the methodology section. For example, in Sec. 3.2 the use of AV-S2UM on predicting both acoustic and semantics discrete speech units is not very clear. In addition, in Sec. 3.3 the reader is assumed to be familiar with the RVQ method when you mention the use of 8 discrete units.

2. Missing key details for reproducibility (see questions).

3. Ablation study is missing. It would be interesting to show how each component contributes to the model’s performance (e.g. U_s and U_a units).

4. Experimental results does not include all state-of-the-art models for all modalities. For example, the authors could add comparisons with DiffGesture and Audio2Gestures for gesture evaluation.

**Questions:**

1. How robust is the AV-S2UM module if one modality (speech or visual) is (partially) missing?

2. How good is your approach in handling variations in gesture expressiveness, especially for different speech prosody?

3. How long can your approach generate frames continuously without degradation in quality? Are there any limitations on sequence length, and can you provide examples for different animation durations (e.g., 3, 6, and 10 seconds)?

4. Can you provide a table comparing inference speeds across methods for a certain input length (e.g. 10 sec) on common hardware along with the number of parameters for each model?

5. Table 3: You compare your approach with ReVISE on LSR3 dataset which was also used to train your model. This raises a question about the generalizability of your model to other environmental conditions. Using different (more diverse) datasets to evaluate your approach could help show its generalization capabilities.

6. Missing details

        1. Sec 3.3: What is the face decoder you are using and how is it combined with the f_s? How do you obtain the f_s feature (eg which architecture is used)?

        2. Sec. 3.3: what is the architecture of the discriminator D and generator G you are using?

        3. Sec. 3.3: what exactly are the functions of the reconstruction loss and the synchronization loss?

        4. Sec 3.3: what are the losses for other modalities like gesture?

---

### Note · Authors · 2024-11-14

I have read and agree with the venue's withdrawal policy on behalf of myself and my co-authors.